# Model Checking with Right Censored Data Using Relative Belief Ratio

**DOI:** 10.3390/e24111579

**Published:** 2022-10-31

**Authors:** Luai Al-Labadi, Ayman Alzaatreh, Mark Asuncion

**Affiliations:** 1Department of Mathematical & Computational Sciences, University of Toronto Mississauga, Mississauga, ON L5L 1C6, Canada; 2Department of Mathematics and Statistics, American University of Sharjah, Sharjah 61174, United Arab Emirates

**Keywords:** beta-Stacy process, model checking, nonparametric Bayesian statistics, relative belief inferences, right-censored data

## Abstract

Model checking is a topic of special interest in statistics. When data are censored, the problem becomes more difficult. This paper employs the relative belief ratio and the beta-Stacy process to develop a method for model checking in the presence of right-censored data. The proposed method for the given model of interest compares the concentration of the posterior distribution to the concentration of the prior distribution using a relative belief ratio. We propose a computational algorithm for the method and then illustrate the method through several data analysis examples.

## 1. Introduction

An important problem in statistics is to check whether a chosen statistical model is in agreement with observed data. This problem is known as *model checking.* The tests used to describe how well a statistical model fits a set of observations are *goodness-of-fit tests*. If it is determined that the observed data do not contradict the model, the true values of the parameters can be inferred. If the model fails to pass its checks, there is cause for concern about the accuracy of the inferences used in analysis. Thus, checking a proposed model on the basis of observed data is a critical component of statistical inference.

The time until an event occurs is the variable of interest in survival analysis. It is common to refer to this time as the *survival time*. In this case, the observed data are typically right censored. That is, at the right side, the exact survival time can become incomplete, for example, when a subject leaves the study before an event occurs, or when the study ends before the event occurs, right censoring occurs. This type of data are common in medical studies where the variable of interest is the time to death or time to relapse of a disease, in reliability studies where the variable of interest is the time until a machine part fails, and in social sciences where the variable of interest is the lifetime of the elderly in certain social programs [1]. While there are several well-known methods for model checking, the approach we want to take is Bayesian in nature. There is interest in developing nonparametric Bayesian procedures for hypothesis testing. The majority of this work focuses on uncensored data. The proposed model was embedded as a null hypothesis into a larger family of distributions, which is the main approach of model testing. Then priors were placed on the null and the alternative, and a Bayes factor was computed (see, for example, [2,3,4,5,6]). A second important approach for model testing is to apply a prior to the true distribution generating the data, and then measure the distance between the posterior distribution and the hypothesized distribution (see, for example, [7,8,9]). A third and more recent approach combines the second approach with relative belief ratio ([10]) (for instance, see the work of [11,12,13]).

When data are censored, goodness-of-fit tests become a challenging problem. This explains why there are few and scattered studies on goodness-of-fit tests for right-censored data. In particular, Bayesian goodness-of-fit tests are rarely covered. Some exceptions include the work of [14,15,16] considered the two sample problem by computing the Bayesian factors and beta process priors ([17]). Al-Labadi and Zarepour [15] used the beta-Stacy process and the Kolmogorov–Smirnov distance to test simple hypotheses. Chen and Hanson [16] used Polya tree priors and computed the Bayesian factor. To compute the posteriors, the MCMC procedure was used. Permutation tests were used to compute *p*-values. These approaches are computationally intensive.

The main problem addressed in this paper is model checking for lifetime distribution functions. A general description of the approach developed in this paper is provided along with a discussion of the benefits that the approach brings to the problem of model checking. The beta-Stacy process $BSP(k\left(t\right),F_{0}\left(t\right))$, t≥0 is considered as a prior on on the space of cumulative distribution functions ([18]). Here, k(·)>0 is a function defined on $R^{+}$, and $F_{0}$ is a cumulative distribution function (cdf) on $R^{+}=\left[0,\infty \right)$ and it represents the prior guess for the beta-Stacy process. How closely realizations from the process are clustered around $F_{0}$ is determined by the value of k(·). The use of the beta-Stacy process is justified because, unlike the Dirichlet process, it is conjugate to both exact and right-censored observations ([19]). The proposed method then compares the posterior and prior distribution concentrations about the model of interest. If the posterior is more concentrated on the model than the prior, this is evidence in favor of the model; if the posterior is less concentrated, this is evidence against the model. This comparison is produced using a relative belief ratio that measures the evidence in the observed data for or against the model and provides a measure of the strength of this evidence; thus, the methodology is based on a direct measure of statistical evidence. The approach is fairly simple to implement and does not necessitate obtaining a closed form of the relative belief ratio. Theoretical results are developed to support appropriate hyperparameter selections k(·) and $F_{0}$.

The rest of the paper is structured as follows. Section 2 provides the background on the definitions and generic properties of the beta-Stacy process, the relative belief ratio, and the Cramér–von Mises distance between probability measures. Section 3 discusses the proposed methodology and the selection of relevant values for the beta-Stacy process’s hyperparameters. Section 4 develops a computational algorithm for implementing the approach. Section 5 examines the performance of the methodology through a number of examples. This method works well in general and can be used for both censored and uncensored observations. Section 6 wraps up the paper with a summary of the findings. Lastly, datasets and figures are provided in the Appendix A.

## 2. Notations and Background

Let the random sample $X_{1},\ldots ,X_{n}$ be drawn from an unknown cdf *F* defined on $R^{+}$. Let $\left(T,\delta \right)=\left(T_{1},\delta_{1}\right),\ldots ,(T_{n},$
$\delta_{n})$ be observed, where $T_{i}=\min\left(X_{i},C_{i}\right)$, $\delta_{i}=I\left(X_{i}\leq C_{i}\right)$ and $C_{1},\ldots ,C_{n}$ are the censoring times. When $\delta_{i}=1,$
$X_{i}$ is observed, and when $\delta_{i}=0,$
$X_{i}$ is right censored, this type of data can occur, for instance, when $X_{i}$ is the lifetime of a patient who enrols in a study of a certain disease. If death occurs from the disease before the study ends, $X_{i}$ is recorded; however, if the patient is still alive, withdraws, or dies for another reason at the end of the study, $C_{i}$ records the end of the period during which the patient was under observation ([20]).

In survival analysis, cdf *F* is frequently referred to as the lifetime distribution function. We use the same notation throughout this paper for the probability measure and its corresponding cumulative distribution function, i.e., F(t)=F((−∞,t]).

### 2.1. Beta-Stacy Process

Walker and Muliere [18] introduced the beta-Stacy process, which is a nonparametric prior that is widely used in survival analysis. In this section, a relevant introduction about the beta-Stacy process is provided.

The beta-Stacy process is defined by another process known as the *log-beta* process ([18]). Let β(·) be a positive function defined on $R^{+}$, and α be a measure concentrated on $R^{+}$ that is absolutely continuous with respect to the Lebesgue measure such that $\int_{0}^{\infty }{\left(\beta (z)\right)}^{-1}\alpha (dz)=\infty .$ On the basis of [18], ${\left\{Z(t)\right\}}_{t\geq 0}$ is a *log-beta process* with parameters α and β, written as Z(t)∼logBSP(α(t),β(t)), if ${\left\{Z(t)\right\}}_{t\geq 0}$ is a Lévy process with Lévy measure defined, for s>0, by
$$L_{t}\left(ds\right)= [\frac{1}{1-e^{-s}}\int_{0}^{t}e^{-s\beta (z)}\alpha \left(dz\right)]ds$$
and has the following moment generating function: $$\log Ee^{-uZ_{c}\left(t\right)}=\int_{0}^{\infty }\left(e^{-us}-1\right)L_{t}\left(ds\right),\;t\geq 0,u\in R.$$

In order to define the beta-Stacy process, let positive function k(·) be defined on $R^{+}$ and the absolutely continuous cdf $F_{0}$ be defined on $R^{+}$. We call ${\left\{F(t)\right\}}_{t\geq 0}$ a beta-Stacy process with parameters k(·) and $F_{0}$, written as $F\left(t\right)∼BSP(a\left(t\right),F_{0}\left(t\right))$, if $F\left(t\right)=1-e^{-Z(t)}$, and ${\left\{Z(t)\right\}}_{t\geq 0}$ is a log-beta process with parameters
$$\alpha \left(dz\right)=k\left(z\right)F_{0}\left(dz\right)\;\;\mathrm{and}\;\;\beta \left(z\right)=k\left(z\right)F_{0}\left(\left[z,\infty \right)\right).$$

Walker and Muliere [18] showed that
(1)$$E [F(t)]=F_{0}\left(t\right)=1-\exp\left(-\int_{0}^{t}{\left(\beta (z)\right)}^{-1}\alpha (dz)\right),$$
rendering $F_{0}$ the prior guess. The expression in (Equation 1) explains the need for assumption $\int_{0}^{\infty }{\left(\beta (z)\right)}^{-1}\alpha (dz)=\infty$. The beta-Stacy process includes various neutral-to-right processes proposed in the literature. For instance, the beta-Stacy process reduces to the Dirichlet process when k(t)=k>0 for all t≥0. The beta-Stacy process also reduces to a simple homogenous process ([21]) when β(·) is constant.

Next, we describe the posterior distributions of ${\left\{Z(t)\right\}}_{t\geq 0}$ and ${\left\{F(t)\right\}}_{t\geq 0}$. Let random sample $X_{1},\ldots ,X_{n}$ be drawn from *F*, and $\left(T,\delta \right)=\left(T_{1},\delta_{1}\right),\ldots ,\left(T_{n},\delta_{n}\right)$ be observed, where $T_{i}=\min\left(X_{i},C_{i}\right),$
$\delta_{i}=I\left(X_{i}\leq C_{i}\right)$ and $C_{1},\ldots ,C_{n}$ are censoring times. Define the *counting process N* by
$$N\left(t\right)=\sum_{i=1}^{n}I\left(T_{i}\leq t\;\mathrm{and}\;\delta_{i}=1\right)$$
and the (left-continuous) *at-risk process Y* by
$$Y\left(t\right)=\sum_{i=1}^{n}I\left(T_{i}\geq t\right),$$
where *I* is the indicator function. In particular, let $N\left\{t\right\}=N\left(t\right)-N\left(t^{-}\right)$ (i.e., N{t} is the number of observed $X_{i}$’s at the exact position *t*). Let ${\left\{Z(t)\right\}}_{t\geq 0}∼\log BSP\left(\alpha \left(t\right),\beta \left(t\right)\right)$ and $F\left(t\right)=1-\exp\left(-Z(t)\right).$ Given the data (T,δ), the posterior distribution of *Z* is a log-beta process ([18]) with parameters
(2)$$\alpha^{*}\left(t\right)=\alpha \left(t\right)+N\left(t\right)$$
and
(3)$$\beta^{*}\left(t\right)=\beta \left(t\right)+Y\left(t\right)-N\left\{t\right\}.$$

The posterior Lévy measure for Z(t) is given by
$$L_{t}^{☆}\left(ds\right)= [\frac{1}{1-e^{-s}}\int_{0}^{t}e^{-s\left(\beta (z)+Y(z)\right)}d\alpha \left(z\right)]ds.$$

There are fixed points of discontinuity in the posterior process. These extra points appear at the exact (uncensored) observations. If $t_{i}$ is an exact observation with corresponding jump $S_{i}$, then
(4)$$1-\exp\left(-S_{i}\right)∼\mathrm{beta}\left(N\left\{t_{i}\right\},\beta \left(t_{i}\right)+Y\left(t_{i}\right)-N\left\{t_{i}\right\}\right).$$

If $N\{t_{i}\}=1,$ then the random jump $S_{i}$ follows an exponential density with mean ${ [\beta \left(t_{i}\right)+Y\left(t_{i}\right)-N\left\{t_{i}\right\}]}^{-1}.$

Let $F\left(t\right)∼BSP\left(k\left(t\right),F_{0}\left(t\right)\right)$, t≥0. Given the data (T,δ), the posterior distribution of *F* is a beta-Stacy process with parameters
(5)$$\begin{array}{c} k^{*}\left(t\right)=\frac{\beta^{*}\left(t\right)}{F_{0}^{*}\left[t,\infty \right)}=\frac{\beta (t)+Y(t)-N\{t\}}{F_{0}^{*}\left[t,\infty \right)} \end{array}$$
and
(6)$$\begin{array}{c} F_{0}^{*}\left(t\right)=1-\prod_{\left[0,t\right]}\left(1-\frac{d\alpha^{*}\left(z\right)}{\beta^{*}\left(z\right)+\alpha^{*}\left\{z\right\}}\right)=1-\prod_{\left[0,t\right]}\left(1-\frac{k\left(z\right)dF_{0}\left(z\right)+dN\left(z\right)}{k\left(z\right)F_{0}\left[z,\infty \right)+Y\left(z\right)}\right), \end{array}$$
with $\alpha^{*}\left(s\right)$ and $\beta^{*}\left(s\right)$ are defined in (Equation 2) and (Equation 3), and ∏ stands for the product integral. Note that, as k(·) tends to zero, the nonparametric Kaplan–Meier estimator of the distribution function is obtained. On the other hand, $F^{*}$ becomes the prior guess $F_{0}$ as k(·) grows large. Thus, parameter k(t) can be viewed as the concentration parameter. The posterior consistency of the beta-Stacy process was addressed by Kim and Lee [22].

Algorithms A and B in Appendix A can be used to sample from prior and posterior beta-Stacy processes. These algorithms were developed by Al-Labadi and Zarepour [15] (see also Lee and Kim [23]).

### 2.2. Relative Belief Ratio

Evans’ (2015) relative belief ratio has become a widely used tool in statistical hypothesis testing theory. Assume that we have a statistical model defined by the pdf $\left\{f_{\theta }:\theta \in \Theta \right\}$ with respect to the Lebesgue measure on the parameter space Θ. Let π(θ) to be a prior on Θ. After observing the data (T,δ), the posterior distribution of θ is given by the conditional density function
$$\pi \left(\theta |(T,\delta )\right)=\frac{f_{\theta }\left((T,\delta )\right)\pi \left(\theta \right)}{\int_{\Theta }f_{\theta }\left((T,\delta )\right)\pi \left(\theta \right)d\theta },$$

Assume that we are interested in inferring regarding the parameter θ. Let π and π(·|(T,δ)) be continuous at θ. Then, the relative belief ratio for a hypothesized value $\theta_{0}$ of θ is given by
(7)$$RB_{\Theta }\left(\theta_{0}\;|\;\left(T,\delta \right)\right)=\pi \left(\theta_{0}\;|\;\left(T,\delta \right)\right)/\pi \left(\theta_{0}\right),$$

The posterior density to the prior density ratio at $\theta_{0},$ that is, $RB(\theta_{0}\;|\;\left(T,\delta \right)),$ measures how beliefs about $\theta_{0}$ changed from a priori to a posteriori. When π and π(·|(T,δ)) are discrete, the relative belief ratio is defined through limits. For more information, see Evans (2015).

Quantity $RB_{\Theta }\left(\theta_{0}\;|\;\left(T,\delta \right)\right)$ is a measure of evidence that $\theta_{0}$ is the true value. If $RB_{\Theta }\left(\theta_{0}\;|\;\left(T,\delta \right)\right)>1$, then the probability of $\theta_{0}$ being the true value increases from a priori to a posteriori; thus, there is evidence based on the data that $\theta_{0}$ is the true value, and there is, hence, evidence in favor of $\theta_{0}$. If $RB_{\Theta }\left(\theta_{0}\;|\;\left(T,\delta \right)\right)<1$; then, the probability of $\theta_{0}$ being the true value decreases from a priori to a posteriori. As a result, the data provide evidence that $\theta_{0}$ is not the true value. Case $RB_{\Theta }\left(\theta_{0}\;|\;\left(T,\delta \right)\right)=1$ implies there is no evidence in either direction.

The ability to calibrate relative belief ratios is an appealing feature that renders it desirable in hypothesis testing problems. After calculating the relative belief ratio, it is critical to determine whether the result represents strong or weak evidence for or against $H_{0}:\theta =\theta_{0}$. A typical calibration $RB(\theta_{0}\;|\;\left(T,\delta \right))$ is obtained by computing the tail probability (Evans, 2015)
(8)$$Str_{\Theta }\left(\theta_{0}|\left(T,\delta \right)\right)=\Pi \left(RB_{\Theta }\left(\theta \;|\;\left(T,\delta \right)\right)\leq RB_{\Theta }\left(\theta_{0}\;|\;\left(T,\delta \right)\right)\;|\;\left(T,\delta \right)\right),$$
where Π(·|(T,δ)) is the posterior distribution of the posterior density π(·|(T,δ)). The posterior probability that the true value of θ has a relative belief ratio no greater than that of the hypothesized value $\theta_{0}$ is represented by Equation (Equation 8). When $RB_{\Theta }\left(\theta_{0}\;|\;\left(T,\delta \right)\right)<1,$ there is evidence against $\theta_{0},$ and a small value for $Str_{\Theta }\left(\theta_{0}|\left(T,\delta \right)\right)$ represents a large posterior probability that the true value has a relative belief ratio greater than $RB_{\Theta }\left(\theta_{0}\;|\;\left(T,\delta \right)\right)$ and hence there is strong evidence against $\theta_{0}.$ Similarly when $RB_{\Theta }\left(\theta_{0}\;|\;\left(T,\delta \right)\right)>1,$ indicates a strong evidence in favour of $\theta_{0},$ while a small value of $Str_{\Theta }\left(\theta_{0}|\left(T,\delta \right)\right)$ indicates weak evidence in favour of $\theta_{0}.$ Figure 1 illustrates the strength of the evidence for both cases; $RB_{\Theta }\left(\theta_{0}\;|\;\left(T,\delta \right)\right)<1$ and $RB_{\Theta }\left(\theta_{0}\;|\;\left(T,\delta \right)\right)>1.$

### 2.3. Cramér–Von Mises Distance

Let *F* and *G* be two cdfs; Cramér–von Mises distance between *F* and *G* is defined as
$$d\left(F,G\right)=\int_{-\infty }^{\infty }{\left(F(x)-G(x)\right)}^{2}G\left(dx\right).$$

Other distances can be used (see Gibbs and Su (2002)), but *d* has some computational advantages. The formula for the distance between the beta-Stacy process and a continuous cdf is provided in the following result.

**Lemma 1.** 
*Let $F_{0}$ be a continuous cdf and $F_{\epsilon }=1-\exp\left(-Z_{\epsilon }\left(t\right)\right)$, where $Z_{\epsilon }\left(t\right)=\sum_{i=1}^{M}J_{i}\delta_{\theta_{i}}\left( [0,t]\right)$, where t≥0 with $\theta_{1},\ldots ,\theta_{M}\overset{i.i.d.}{∼}F_{0}$. Let $\theta_{\left(1\right)}\leq \ldots \leq \theta_{\left(M\right)}$ denote the order statistics of $\theta_{1},\ldots ,\theta_{M}$ and $J_{1}^{\prime },\cdots ,J_{M}^{\prime }$ be the associated jump sizes such that $J_{i}=J_{j}^{\prime }$ when $\theta_{i}=\theta_{\left(j\right)}.$ Then*

$$\begin{array}{ccc} d(F_{\epsilon },F_{0}) & = & \frac{1}{3}-\sum_{i=0}^{M}F_{\epsilon }\left(\theta_{\left(i\right)}\right) [{\left(F_{0}\left(\theta_{\left(i+1\right)}\right)\right)}^{2}-{\left(F_{0}\left(\theta_{\left(i\right)}\right)\right)}^{2}]\\ & & +\sum_{i=0}^{M}{\left(F_{\epsilon }\left(\theta_{\left(i\right)}\right)\right)}^{2} [F_{0}\left(\theta_{\left(i+1\right)}\right)-F_{0}\left(\theta_{\left(i\right)}\right)]. \end{array}$$

*where $F_{\epsilon }\left(\theta_{\left(i\right)}\right)=1-\exp\left(-\sum_{k=1}^{i}J_{k}^{\prime }\right)$.*


**Proof.** Note that
$$F_{\epsilon }\left(x\right)=\left\{\begin{array}{cc} 0 & \mathrm{if}\;x<\theta_{\left(1\right)}\\ F_{\epsilon }\left(\theta_{\left(i\right)}\right) & \mathrm{if}\;\theta_{\left(i\right)}\leq x<\theta_{\left(i+1\right)}\;\left(i=1,\ldots ,M-1\right).\\ 1 & \mathrm{if}\;x\geq \theta_{\left(M\right)} \end{array}\right.$$Le $\theta_{\left(0\right)}=0$ and $\theta_{\left(M+1\right)}=+\infty .$ Then,
$$\begin{array}{ccc} d(F_{\epsilon },F_{0}) & = & \int_{\theta_{\left(0\right)}}^{\theta_{\left(\epsilon +1\right)}}{ [F_{\epsilon }\left(x\right)-F_{0}\left(x\right)]}^{2}f_{0}\left(x\right)dx\\ & = & \sum_{i=0}^{M}\int_{\theta_{\left(i\right)}}^{\theta_{\left(i+1\right)}}{ [F_{\epsilon }\left(\theta_{\left(i\right)}\right)-F_{0}\left(x\right)]}^{2}f_{0}\left(x\right)dx. \end{array}$$Substituting $y=F_{0}\left(x\right)$ and $U_{\left(i\right)}=F_{0}\left(\theta_{\left(i\right)}\right)$ gives
$$\begin{array}{ccc} d(F_{\epsilon },F_{0}) & = & \sum_{i=0}^{M}\int_{U_{\left(i\right)}}^{U_{\left(i+1\right)}}{ [F_{\epsilon }\left(\theta_{\left(i\right)}\right)-y]}^{2}dy\\ & = & \frac{1}{3}\sum_{i=0}^{M}\left\{{\left[F_{\epsilon }\left(\theta_{\left(i\right)}\right)-U_{\left(i\right)}\right]}^{3}-{\left[F_{\epsilon }\left(\theta_{\left(i\right)}\right)-U_{\left(i+1\right)}\right]}^{3}\right\}.\\ & = & \frac{1}{3}\sum_{i=0}^{M} [U_{\left(i+1\right)}^{3}-U_{\left(i\right)}^{3}]-\sum_{i=0}^{M}F_{\epsilon }\left(\theta_{\left(i\right)}\right) [U_{\left(i+1\right)}^{2}-U_{\left(i\right)}^{2}]\\ & & +\sum_{i=0}^{M}F_{\epsilon }^{2}\left(\theta_{\left(i\right)}\right) [U_{\left(i+1\right)}-U_{\left(i\right)}]\\ & = & \frac{1}{3}-\sum_{i=0}^{M}F_{\epsilon }\left(\theta_{\left(i\right)}\right) [U_{\left(i+1\right)}^{2}-U_{\left(i\right)}^{2}]+\sum_{i=0}^{M}(\left(F_{\epsilon }\right){\left(\theta_{\left(i\right)}\right)}^{2} [U_{\left(i+1\right)}-U_{\left(i\right)}]\\ & = & \frac{1}{3}-\sum_{i=0}^{M}F_{\epsilon }\left(\theta_{\left(i\right)}\right) [{\left(F_{0}\left(\theta_{\left(i+1\right)}\right)\right)}^{2}-{\left(F_{0}\left(\theta_{\left(i\right)}\right)\right)}^{2}]\\ & & +\sum_{i=0}^{M}{\left(F_{\epsilon }\theta_{\left(i\right)}\right)}^{2} [F_{0}\left(\theta_{\left(i+1\right)}\right)-F_{0}\left(\theta_{\left(i\right)}\right)]. \end{array}$$

□

When considering the prior and posterior distributions of the Cramér–von Mises distance, the following lemma allows for the use of the approximation to $BSP(k\left(\cdot \right),F_{0})$.

**Lemma 2.** 
*If $F∼BSP(k\left(\cdot \right),F_{0})$ and $F_{\epsilon }$ is given by Algorithm A, then $d\left(F_{\epsilon },F_{0}\right)\overset{a.s.}{\to }d\left(F,F_{0}\right)$ as ϵ→0.*


**Proof.** By Kim and Lee (2001), $F_{\epsilon }\left(x\right)\overset{a.s.}{\to }F\left(x\right)$, where a.s. stands for almost surely convergence. As ${\left(F_{\epsilon }\left(x\right)-F_{0}\left(x\right)\right)}^{2}\leq f_{0}\left(x\right)$, where $f_{0}\left(x\right)$ is integrable, the proof follows from the dominated convergence theorem. □

## 3. Model Checking Using the Relative Belief

Consider the statistical model $\left\{F_{\theta }:\theta \in \Theta \right\}$ of continuous cdf on $R^{+}$. Let $X_{1},\ldots ,$
$X_{n}$ be a random sample drawn from an unknown cdf *F* defined on $R^{+}$. Assume that $\left(T,\delta \right)=(\left(T_{1},\delta_{1}\right),\ldots ,\left(T_{n},\delta_{n}\right))$ is observed, where $T_{i}=\min\left(X_{i},C_{i}\right),$
$\delta_{i}=I\left(X_{i}\leq C_{i}\right)$ and $C_{1},\ldots ,C_{n}$ are censoring times. The aim is to test the hypothesis $H_{0}:F\in \left\{F_{\theta }:\theta \in \Theta \right\}$. Let $BSP(k\left(\cdot \right),F_{0})$ be the prior on *F* for some choice of k(·) and $F_{0}$. Then $F\left(t\right)\;|\;\left(T,\delta \right)∼BSP\left(k^{*}\left(t\right),F_{0}^{*}\left(t\right)\right)$, where k(·) and $F_{0}$ are defined in (Equation 5) and (Equation 6), respectively. If $H_{0}$ is true, then the posterior distribution of the distance between *F* and the proposed model should be more concentrated around 0 than the prior distribution. As a result, this test involves comparing the concentrations of the prior and the posterior distributions of *d* (see Lemma 1) about 0 using the relative belief ratio with the interpretation as discussed in Section 2.2.

To perform this test, we must measure the distance and then set relevant values for k(·) and $F_{0}$. To calculate the distance, similar to Al-Labadi and Evans [24], we computed $d(F,F_{\hat{\theta }})$, where $\hat{\theta }$ is the relative belief estimate of θ, which is always the same as the maximal likelihood estimate (MLE) for the full model parameter. In terms of hyperparameters, we set $F_{0}=F_{\hat{\theta }}$ and k(t)=k for all *t*. There are numerous advantages in setting $F_{0}=F_{\hat{\theta }}$. First, it avoids *prior-data conflict*, a possible contradiction between the data and the prior. This typically happens when the prior places its mass in a region of the parameter space where the data suggest the true value does not lie ([25,26]). In the context of the approach considered in this paper, prior-data conflict arises whenever there is a small overlap between the effective support regions of *F* and $F_{\hat{\theta }}$. Note that, by Lemma 1, the distance $d(F,F_{\hat{\theta }})$ depends on the prior guess $F_{0}$ through the jump points $\theta_{i}$. If the $\theta_{i}$ lay in one tail of $F_{\hat{\theta }}$, then we get prior-data conflict between *F* and $F_{\hat{\theta }}$ because $F_{0}$ and *F* had the same effective support. To avoid this, it is required that $\theta_{i}$ are selected in a region that includes most of the mass of $F_{\hat{\theta }}$. When $F_{0}=F_{\hat{\theta }}$, then $F_{\hat{\theta }}$ is the prior mean of *F*; thus, both share the same effective support, which renders it a reasonable choice to avoid prior-data conflict. We refer the reader to Example 1 of Al Labadi and Evans (2018) for an interesting discussion about prior-data conflict. Nevertheless, the choice of $F_{0}=F_{\hat{\theta }}$ should also avoid any impacts due to the “double use of the data”. This means that the approach becomes conservative in detecting the model failure when $H_{0}$ is false. Although setting $F_{0}=F_{\hat{\theta }}$ appears to induce a data-dependent prior distribution for *d*, the following lemma implies that this is not the case; thus, the approach is prior distribution-free with this choice.

**Lemma 3.** 
*If $F∼BSP\left(k\left(\cdot \right),F_{\hat{\theta }}\right)$, then the distribution of $d\left(F,F_{\hat{\theta }}\right)$ does not depend on $F_{\hat{\theta }}$.*


**Proof.** Using Lemma 1, since ${\left(\theta_{i}\right)}_{1\leq i\leq M}$ is a sequence of i.i.d. random variables with continuous distribution $F_{\hat{\theta }},$ for i≥1, we have $U_{i}\overset{d}{=}F_{\hat{\theta }}\left(\theta_{i}\right)$, where ${\left(U_{i}\right)}_{1\leq i\leq M}$ is a sequence of i.i.d. random variables follow a uniform distribution on [0,1]. Thus,
$$\begin{array}{ccc} d(F_{\epsilon },F_{\hat{\theta }}) & \overset{d}{=} & \frac{1}{3}-\sum_{i=0}^{M}F_{\epsilon }\left(\theta_{\left(i\right)}\right) [{\left(U\left(i+1\right)\right)}^{2}-{\left(\theta_{\left(i\right)}\right)}^{2}]\\ & & +\sum_{i=0}^{M}{\left(F_{\epsilon }\left(\theta_{\left(i\right)}\right)\right)}^{2} [U_{\left(i+1\right)}-U_{\left(i\right)}], \end{array}$$
where $U_{\left(i\right)}$ is the *i*-th order statistic for ${\left(U_{i}\right)}_{1\leq i\leq M}$ i.i.d. uniform[0,1]. Now, as ϵ→0, by Lemma 2, we conclude that the distribution of $d(F,F_{\hat{\theta }})$ does not depend on $F_{\hat{\theta }}$. □

The following results shows that setting $F_{0}=F_{\hat{\theta }}$ prevents any effect due to the double use of the data. Specifically, as the sample size increases, the posterior distribution of $d\left(F,F_{\hat{\theta }}\right)$ becomes concentrated around 0 if and only if $H_{0}$ is true. For the proof, see Al-Labadi and Evans (2018).

**Lemma 4.** 
*Let $\left(T,\delta \right)=(\left(T_{1},\delta_{1}\right),\ldots ,\left(T_{n},\delta_{n}\right))∼F$, where $F∼BSP\left(k\left(\cdot \right),F_{\hat{\theta }}\right)$. Suppose that $\hat{\theta }\overset{a.s.}{\to }\theta_{0},\sup_{y}\left|F_{\hat{\theta }}\left(y\right)-F_{\theta_{0}}\left(y\right)\right|\overset{a.s.}{\to }0$ as n→∞.*
*(i)* 
*If $H_{0}$ is true, then, as n→∞, $d\left(F|\left(T,\delta \right),F_{\hat{\theta }}\right)\overset{a.s.}{\to }0$.*
*(ii)* 
*If $H_{0}$ is false, then, as n→∞, $\lim\inf d\left(F|\left(T,\delta \right),F_{\hat{\theta }}\right)\overset{a.s.}{>}0.$*



Now, concerning the choice of k(·), we considered k(t)=k for all t>0. In general, larger values of *k* must be chosen to identify smaller deviations. Consequently, it is possible to consider multiple values of *k*. One way to perform that is, for instance, to start with k=1. If a larger (smaller) value of *k* renders the relative belief ratios to be below (above) 1, $H_{0}$ is rejected (accepted). As Section 5 shows, when the null hypothesis is correct (not correct), the relative belief ratio always remains above (below) 1 when larger (smaller) values of *k* are considered. When using the Dirichlet process, Al-Labadi and Zarepour [27] advised using k≤0.5n for complete data to avoid the prior becoming too influential. Setting *k* between 1 and 10 is satisfactory for most purposes. As indicated in the introduction, when k(t)=k, the beta-Stacy process turns into the Dirichlet process. However, in the presence of right censored data, the posterior distribution of the Dirichlet process becomes beta-Stacy process ([19]). This justifies the necessity of using the beta-Stacy process in the approach.

## 4. Computational Algorithm

Closed forms of the prior and posterior densities of $D=d(F,F_{\hat{\theta }})$ are required to compute the relative belief ratio as in (Equation 7). This is not usually available. As a result, the relative belief ratio must be approximated through simulation. A particular problem of computing (Equation 7) arises here when both $\pi_{D}\left(0\right)$ and $\pi_{D|(T,\delta )}\left(0\right)$ are close to 0, where $\pi_{D}\left(\cdot \right)$ and $\pi_{D|(T,\delta )}\left(\cdot \right)$ denote the pdf’s of *D* and D|(T,δ), respectively. In such a case, determining $RB_{D}\left(0\;|\;\left(T,\delta \right)\right)$ is difficult. The formal definition of the relative belief ratio, as discussed in Section 2.2, is as a limit that can be approximated at zero by
(9)$$\begin{array}{c} {\hat{RB}}_{D}\left(0\;|\;T,\delta \right)=\frac{\Pi_{D}\left(\left[0,d_{c}\right)\right)}{\Pi_{D|(T,\delta )}\left(\left[0,d_{c}\right)\right)}, \end{array}$$
for a suitably small value $d_{c}$, where $\Pi_{D}\left(\cdot \right)$ and $\Pi_{D|(T,\delta )}\left(\cdot \right)$ denote the cdfs of *D* and D|(T,δ), respectively. From Equation (Equation 8), the strength of the evidence based on the relative belief ratio $RB_{D}\left(0\;|\;T,\delta \right)$ can be computed using
(10)$$\begin{array}{c} Str_{D}\left(0\;|\;\left(T,\delta \right)\right)=\Pi_{D|(T,\delta )}\left(RB_{D}\left(d\;|\;\left(T,\delta \right)\right)\leq RB_{D}\left(0\;|\;\left(T,\delta \right)\right)\right). \end{array}$$

Appendix B provides computational Algorithm C for assessing $H_{0}$ on the basis of estimates of (Equation 9) and (Equation 10). A similar algorithm for complete data based on the Dirichlet process was developed by Al-Labadi and Evans [24].

## 5. Examples

The approach is demonstrated by two main examples in this section. Throughout this section, let Exp (λ), Weibull (k,λ), and Lognorma l (μ,σ) denote the exponential distribution with mean 1/λ, the Weibull distribution with shape parameter *k* and scale parameter λ, and the log-normal distribution with mean μ and standard deviation σ, respectively. In all examples, the sensitivity to choosing *k* is investigated. We set ϵ=0.01, $i_{0}=1$, $M_{0}=20$, and $r_{1}=r_{2}=1000$ in Algorithms A, B, and C, though other values are also possible. R package **parmsurvfit** was used to compute the MLE of the distribution parameters.

**Example 1.** 
*A real dataset from Lee and Wang (2003) based on the remission times (in months) of cancer patients. The dataset is given in Appendix C. We tested hypothesis*


$H_{0}:$


*F, given in Table 1, is the underlying distribution of the observed data,*

*where F could be a family of distributions (composite hypothesis) or a specific distribution with known parameters (simple hypothesis). Various values of k were considered to investigate the approach’s sensitivity to concentration parameter selection. The *p*-value of the (frequentist) log-rank test was also computed for comparison purposes. Table 1 summarizes the findings. When $H_{0}$ is true, we want RB>1 and a strength close to 1, and when $H_{0}$ is false, we want RB<1 and a strength close to 0. According to Table 1, the proposed test performed well in this example.*


**Example 2.** 
*Simulated data. The primary purpose of this dataset is to investigate how the proposed test performs as the sample size increases. We considered data $\left(T_{1},\delta_{1}\right),\ldots ,\left(T_{n},\delta_{n}\right)$, where $T_{i}=\min\left(X_{i},C_{i}\right)$ where the survival times ${\left(X_{i}\right)}_{1\leq i\leq n}$ were generated from Lognormal(1,4), while the censored time ${\left(C_{i}\right)}_{1\leq i\leq n}$ are generated from Lognormal (4,1).*


$H_{0}:$


*F in Table 2 is the underlying distribution of the observed data.*


*Table 2 summarizes the results that show that the selected models were accepted. Figure A1, Figure A2 and Figure A3 (see Appendix D) give the plots of $F_{0}=F_{\hat{\theta }}$ and 5 sample paths each for the prior beta-Stacy process and the posterior beta-Stacy process for each case in Table 2. These figures clearly show that the plots of the sample paths for the posterior process moved toward the plot of $F_{0}.$ This supports the previous conclusion regarding the null hypothesis. Furthermore, in this case, the *p*-values of the (frequentist) log-rank test support the conclusion that the null hypothesis should not be rejected.*


## 6. Concluding Remarks

The beta-Stacy process and relative belief ratio were used to propose a general approach for model checking. This method could be used for both complete and right-censored data. It could also be used to test composite or simple hypotheses. Several examples demonstrated that the approach works very well.

Though the Cramér–von Mises distance was used here, other distance measures such as the Kolmogorov–Smirnov and Anderson–Darling distances are viable alternatives. Testing for families of multivariate distributions is an important extension of the approach presented in this paper. While conceptually similar, computational and inferential issues must be addressed. This problem can be addressed in future work.

## Figures and Tables

**Figure 1 entropy-24-01579-f001:**
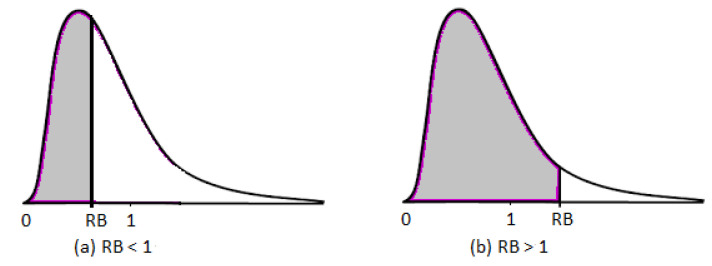
Strength of evidence. The shaded area represents $Str_{\Theta }\left(\theta_{0}|\left(T,\delta \right)\right).$ RB represents the relative belief ratio $RB_{\Theta }\left(\theta_{0}\;|\;\left(T,\delta \right)\right).$ The smaller the shaded area, the stronger the evidence for case (**a**). The larger the shaded area, the stronger the evidence for case (**b**).

**Table 1 entropy-24-01579-t001:** Model checking of Example 1.

*F*	$F_{0}$	*k*	*RB*(*Str*)	*p*-Value
Exp(λ)	Exp$\left(\hat{\lambda }=0.0962\right)$	1	20(1)	0.6521
		5	19.84(1)	
		10	15.98(1)	
Weibull(k,λ)	Weibull$\left(\hat{k}=1.0536,\hat{\lambda }=10.1901\right)$	1	20(1)	0.9986
		5	19.76(1)	
		10	17.58(1)	
Lognormal(μ,σ)	Lognormal$\left(\hat{\mu }=1.8198,\hat{\sigma }=1.0912\right)$	1	20(1)	0.7550
		5	18.48(1)	
		10	15.52 (1)	
Exp(λ=2)	Exp(λ=2)	1	0(0)	0
		5	0(0)	
		10	0(0)	
Weibull(2,0.5)	Weibull(2,0.5)	1	0(0)	0
		5	0(0)	
		10	0(0)	
Lognormal(3,1)	Lognormal(2,1)	1	0(0)	0
		5	0(0)	
		10	0(0)	

**Table 2 entropy-24-01579-t002:** Model checking of Example 2.

*F*	*n*	$F_{0}$	*c*	RB(Str)	*p*-Value
Exp(λ)	20	Exp$\left(\hat{\lambda }=0.03\right)$	1	3.4(0.83)	0.9995
			5	0.76(0.08)	
			10	0.36(0)	
	50	Exp$\left(\hat{\lambda }=0.03\right)$	1	0.24(0.02)	0.9977
			5	0.02(0)	
			10	0(0)	
	100	Exp$\left(\hat{\lambda }=0.05\right)$	1	0.2(0.02)	0.9992
			5	0(0)	
			10	0(0)	
Weibull(k,λ)	20	Weibull$\left(\hat{k}=0.38,\hat{\lambda }=20.88\right)$	1	18.12(1)	1
			5	9.78 (1)	
			10	5.22(0.739)	
	50	Weibull$\left(\hat{k}=0.36,\hat{\lambda }=19.77\right)$	1	19.88(1)	0.9992
			5	15.98(1)	
			10	11.12(1)	
	100	Weibull$\left(\hat{k}=0.38,\hat{\lambda }=14.10\right)$	1	20(1)	0.9991
			5	19.48(1)	
			10	16.46(1)	
Lognormal(μ,σ)	20	Lognormal$\left(\hat{\mu }=1.14,\hat{\sigma }=2.40\right)$	1	11.68(0.416)	0.9867
			5	3.52(1)	
			10	1.88(1)	
	50	Lognormal$\left(\hat{\mu }=1.63,\hat{\sigma }=3.64\right)$	1	20(1)	0.9454
			5	16.4(1)	
			10	12.4(1)	
	100	Lognormal$\left(\hat{\mu }=1.34,\hat{\sigma }=3.44\right)$	1	20(1)	0.9373
			5	19.36 (1)	
			10	16.12(1)	

## Data Availability

Not applicable.

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
