# Peer review of "Model Checking with Right Censored Data Using Relative Belief Ratio"

_entropy, 2022, doi:10.3390/e24111579_

Round 1
Reviewer 1 Report
This manuscript proposes a new and interesting method to evaluate and check the goodness of fit of a relatively simple model for survival analysis, by comparing the concentration of the prior distribution to the posterior distribution under a Beta-Stacy process, using the Relative Belief Ratio.
In the statistics literature, many methods for evaluating the goodness of fit of a model suffer from the "double-data use" problem. For example, the classical fit residual statistics that are used to evaluate the fit of the linear regression model (using ordinary least-squares, OLS) to data
suffer from the double-data use issue because the data are used first to estimate the model parameters via OLS, and the data are used a second time to evaluate the residual fit of the data to the fitted model by estimating it in the first stage from the data. As a result of this double-data use, the residual statistics provide an over-optimistic assessment of the fit of the linear regression model to the data.
Does the Relative Belief Ratio (RBR) also suffer from the double-data use overoptimism issue? I am asking because the data are used to compute the posterior of the beta-Stacy process, and the data are used again to compute the RBR measure of model fit (based on comparing the posterior concentration to the prior concentration). Does the RBR provide an overoptimistic measure of model fit due to using the data twice? The paper needs to address this question somehow.
Also, in Line 263, it is mentioned that the baseline measure (F0) of the beta-Stacy process is set by the maximum likelihood estimate of the maximum likelihood estimate (MLE) of the distribution of the model f interest for testing. Then, is it fair to call the proposed method of goodness of fit testing, proposed in the manuscript, an empirical Bayes method, since a (beta-Stacy) prior baseline parameter is set by the MLE and data? This question should also be addressed in the manuscript.
Details/Suggestions
Abstract, lines 18-25:
The proposed method, for the given model of interest, compares the concentration of the posterior distribution to the concentration of the prior distribution, using a relative belief ratio. We propose a computational algorithm for the method, and then illustrate the method through several data analysis examples.
Line 30: An important problem in statistics is to check ...
Line 42: time can become incomplete, for example, when a subject leaves the study before an event...
Line 68: ...tests become a challenging problem.
Line 88, Line 101, and Line 264: F0 should be F0 (i.e., put the zero in subscript).
Line 89, and Line 101: should k(cdot) be k(.) ?
Line 111: Can safely delete "incorporated".
Line 118: Enter space before the start of the sentence "If the death..."
Line 120: Xi should be Xi. (i.e., put the "i" in subscript).
Line 120: Ci should be Ci. (i.e., put the "i" in subscript).
Line 125: "-infty" should use the proper math symbol for negative infinity.
Line -4 from bottom of page 6: "is a beta-Stacy" --> "a beta-Stacy"
Line 192-193: "Theta" should instead be the proper math symbol for capital theta. And "pi(theta)" should subscript "i" and use the proper math symbol for lowercase theta.
Line 194: by the conditional density function
Line 196: are continuous -> be continuous
Line 199: "theta0" should use the proper math symbol for theta0.
Page 16, Line -5: ration -> ratio
Line 359: Cram'er-von Mises -> Cramer-von Mises.
Line 371,387, 398, 423: Each citation needs to include volume and page numbers. And "Evan" should be "Evans".
Line 429: Delete this line.
Line 434: The Gibbs and Su citation is highly incomplete.
Line 462: "Nuller" should be "Müller".
And should "2237-242" be "237-242"?
Author Response
Kindly see the attached file.
Thanks you!

Reviewer 2 Report
This paper presents a method for model checking with right-censored data, using a Bayesian framework. The relative belief ratio is used to quantify how much closer the posterior is to the supposed model than is the prior; this will be large when the null hypothesis that the model is correct is true and small when false. A strength measure is provided to quantify strength of the evidence.
The paper was clearly written and the idea seems to be sensible. However, there are a number of typographical and notational errors that should be corrected prior to publication. I also have identified several other issues that should be addressed to strengthen the presentation of this work. These are detailed below in rough descending order of importance.
1. The introduction of the beta-Stacy process is interesting. However, the authors go on to specify a constant k (line 288), indicating they are in fact using the special case Dirichlet process. More detail should be provided on why the additional sophistication of the BSP is needed to emphasize the contribution of the work. This is also important as it seems the lead author has done a lot of work in this area already, so what distinguishes the present paper from previous ones (lines 63-67)?
2. The examples need further explanation. It should be clarified that example 1 is based on a real data analysis and example 2 is based on simulations (if this is in fact the case).
a. Why in Table 1 do all the first three hypotheses show strong evidence in favor? It seems impossible that one data set would be generated by three different distributions. The authors should provide a more thorough explanation.
b. Why is the log-rank test a useful point of comparison? Is this just comparing the prior and posterior mean survival curves?
c. Where do the parameter values in the lower part of the table come from and why are these a meaningful point of comparison?
3. In line 337 it is stated that the proposed method may be applied with either composite or simple hypotheses. Does that distinction make any material difference in properties or procedure? Line 251 may be germane: is \theta here meant to be generic or specific?
4. Line 265: could the authors provide more detail about the prior-data conflict mentioned here?
5. Line 220: shouldn’t this read “less than”? A diagram might be helpful here for the reader to understand the relationship between Str and RB, perhaps something like the density curves used to illustrate power for statistical tests. Different cases ({large, small} RB x {large, small} Str) could be presented in different panels.
6. The choice of parameters in line 334 needs more explanation. How are users of this method intended to select these for their own data?
7. Line 270: the distribution-free property seems to derive from the probability integral transform. Is this correct? If so, the authors might elaborate on this to provide more intuition for lemma 3.
8. What is the specific homogeneous process mentioned in line 135?
9. Equation (11): what is the subscript i on the RHS here? The notation could use some tightening up.
10. Another question regarding notation: at the bottom of page 6 the cdf F_0 takes an interval argument; is this meant to correspond to the survival function value at the point z? If so then the interval is not needed, could just write 1-F_0(z).
11. I’m not sure how important it is to have the algorithms A-C presented in-text. I would suggest placing these in supplementary material.
Author Response

(The authors gave the same response as above.)

Round 2
Reviewer 1 Report
I Accept the manuscript in its present form, as it addressed all the comments that I raised on the previous manuscript version.